# Lab-on-a-Chip Devices for Nucleic Acid Analysis in Food Safety

**DOI:** 10.3390/mi15121524

**Published:** 2024-12-21

**Authors:** Inae Lee, Hae-Yeong Kim

**Affiliations:** Institute of Life Sciences & Resources, Department of Food Science and Biotechnology, Kyung Hee University, Yongin 17104, Republic of Korea; ilee3873@khu.ac.kr

**Keywords:** microfluidic, extraction, polymerase chain reaction, loop-mediated isothermal amplification, pathogens

## Abstract

Lab-on-a-chip (LOC) devices have been developed for nucleic acid analysis by integrating complex laboratory functions onto a miniaturized chip, enabling rapid, cost-effective, and highly sensitive on-site testing. This review examines the application of LOC technology in food safety, specifically in the context of nucleic acid-based analyses for detecting pathogens and contaminants. We focus on microfluidic-based LOC devices that optimize nucleic acid extraction and purification on the chip or amplification and detection processes based on isothermal amplification and polymerase chain reaction. We also explore advancements in integrated LOC devices that combine nucleic acid extraction, amplification, and detection processes within a single chip to minimize sample preparation time and enhance testing accuracy. The review concludes with insights into future trends, particularly the development of portable LOC technologies for rapid and efficient nucleic acid testing in food safety.

## 1. Introduction

Lab-on-a-chip (LOC) devices are advanced, miniaturized systems that condense the myriad analytical functions of traditional laboratories into a single, compact chip. Typically, harnessing microfluidics technology, these devices enable intricate chemical and biological analyses using minimal sample volumes [1,2]. LOC chips are designed with micrometer-scale channels and structures that promote the movement, mixing, separation, and analysis of samples [3]. Specifically, microchambers or microchannels within a single chip are designated to perform individual functions, with fluid control relying on either passive methods (e.g., capillary action, gravity, and osmosis) or active systems (e.g., pressure-driven and centrifugal flows) [4]. Passive systems are characterized by their cost-effectiveness, autonomy, and portability, whereas active systems, which incorporate miniaturized mechanical components such as electromagnetics, pumps, and valves, provide enhanced control and reliability throughout multiple operational steps [2]. The primary advantages of LOC technology include substantial time and cost efficiencies resulting from its miniaturized format [5]. By performing experiments with small sample volumes, both material expenses and waste generation are minimized. Additionally, automated LOC systems decrease the potential for errors and expedite analysis times [6]. These devices are particularly well-suited for point-of-care testing (POCT), as their portability allows for on-site diagnostics without the necessity of laboratory facilities. Given these benefits, LOC technology is increasingly being employed in food safety [7,8,9,10], clinical diagnostics [11,12,13], and environmental monitoring [14,15]. For example, a micro-automated microfluidic device was developed for on-site and rapid POCT of *Escherichia coli* (*E. coli*) O157:H7 [10]. The device was connected to a mobile phone to capture RGB data via colorimetry and fluorescence channels, allowing for precise measurement of *E. coli* concentration.

Nucleic acid detection technology is vital to food safety management, serving as an essential tool for the early detection of pathogenic microorganisms and harmful substances, food authenticity verification, and genetically modified organism (GMO) testing [16,17,18,19]. As the food industry continues to globalize, food safety has emerged as an increasingly urgent public health concern, escalating the demand for rapid and accurate detection systems. Traditional methods such as microbial culture or chemical testing are often time-consuming, costly, and may lack precision [7]. In contrast, nucleic acid-based analysis methods offer the ability to swiftly and accurately identify pathogenic microorganisms or specific genetic markers by directly analyzing DNA or RNA, thus providing a transformative approach to food safety management [20]. It facilitates prompt responses to potential risks and plays a critical role in preventing their spread, thereby making substantial contributions to public health and consumer trust [21]. In the food industry, these nucleic acid-based diagnostic technologies are recognized as essential tools for enhancing food safety and reinforcing quality control [22,23].

Nucleic acid analysis typically consists of three main steps: extraction (sample preparation), amplification, and detection. The nucleic acid extraction process involves cell disruption, removal of membrane lipids and proteins, elimination of other nucleic acids, nucleic acid purification, and concentration [24]. This process employs both chemical methods (e.g., osmotic shock, enzymatic digestion, detergents, alkali treatment) and mechanical methods (e.g., heating, homogenization, ultrasonication, pressing, ball milling) [25]. Solid-phase extraction is commonly used to selectively isolate target nucleic acids from solutions via specific hydrophobic, polar, and/or ionic interactions between the solute and the sorbent [25]. Detergents and alkali treatments offer fast, reliable, and simple methods but have low lysis efficiency and moderate DNA purity [26]. Mechanical methods achieve robust lysis but require specialized equipment. Solid-phase extraction, such as silica matrices, provides high-purity DNA and is user-friendly but poses challenges in recovering small DNA fragments and is not reusable [27,28]. The amplification and detection of nucleic acids are carried out using molecular diagnostic technologies, including polymerase chain reaction (PCR) and loop-mediated isothermal amplification (LAMP). These methods amplify minute quantities of DNA or RNA to quickly detect specific pathogens or harmful substances, producing outputs in the form of colorimetric, fluorescent, or electrical signals. Integrating nanomaterials and biosensing technologies into each step of nucleic acid analysis has been widely reported [2,29,30]. These innovations have significantly enhanced the sensitivity and accuracy of nucleic acid analysis while improving portability and user convenience, representing substantial progress in the field. Challenges in high-performance nano-biosensor technology include simultaneously quantifying multiple biomarkers and advancing novel materials, amplification strategies, signal probes, and surface engineering, particularly optimizing surface blocking and washing steps [31,32,33].

Recently, nucleic acid analysis in LOC devices is becoming essential for real-time contamination prevention in food supply chains [8,15,34,35]. LOC devices streamline several preprocessing steps that are critical in traditional food analysis, enhancing food safety and quality control through rapid and precise nucleic acid analysis [7]. For example, LAMP technology utilized in LOC systems can deliver results significantly faster than conventional culture- or PCR-based methods, enabling prompt decision-making [36]. LAMP operates under isothermal conditions (60–65 °C) to produce 10^6^–10^9^ copies of DNA within 30–60 min, outperforming thermal cycling methods, which generate 2^30^ copies in 2–3 h [37]. However, certain limitations in LAMP technology remain, such as primer design complexity and non-specific amplification [38]. This review examines the combination of nucleic acid-based analysis methods with LOC technology. We explore advancements in microfluidic device technologies for nucleic acid extraction, amplification, and detection aimed at identifying foodborne pathogens or contaminants (Figure 1). We also discuss recent technological trends and prospects for integrated LOC devices in the context of real-time, on-site food safety monitoring.

## 2. Innovations in Sample Preparation for Nucleic Acid Analysis

Food samples are complex matrices, necessitating their purification into suitable forms for analysis. A sample preparation process, including nucleic acid extraction and purification, is typically required to analyze microorganisms or genetic markers in food. The pretreatment protocol varies based on sample type (e.g., plant, animal cells, bacteria, and virus) and the intended testing method [39]. Nucleic acid extraction begins with cell lysis to release intracellular material [12]. Chemical lysis typically involves alkaline solutions or surfactants to break down the plasma membrane, with sodium dodecyl sulfate aiding in protein dissolution. Mechanical lysis applies shear stress to physically disrupt the membrane, while thermal lysis uses heat to denature cell membranes. Following extraction, purification removes inhibitors to ensure efficient amplification. Methods include centrifugation, filtration, and magnetic techniques [39]. Centrifugation separates DNA from other components by differential sedimentation, while filtration uses silica membranes to bind DNA at high salt concentrations. Magnetic beads were used to bind target DNA, which is then separated by magnetic force. The pretreatment processes for food testing are typically time-consuming and labor-intensive. However, LOC devices facilitate the integration and automation of multiple pretreatment steps within a miniaturized system, eliminating the need for separate equipment. Microfluidic technology is instrumental in these LOC-based pretreatment systems, as it precisely controls the small volumes of samples and reagents through microchannels, thereby enhancing reaction efficiency and expediting sample processing.

Some LOC platforms are equipped with heating modules designed for the thermal lysis of bacterial cells and DNA extraction. Tsougeni et al. [40] fabricated microfluidic chips on printed circuit boards and poly(methyl methacrylate) substrates featuring a resistive microheater for thermal cell lysis and purification. Thermal lysis was conducted at 94 °C for 13 min, followed by on-chip DNA purification through multiple washing and elution steps. One centrifugal microfluidic LOC system facilitated the thermal lysis of *E. coli* O157:H7 colonies isolated from food samples, along with bacterial DNA amplification and hybridization assays [41]. On-chip cell lysis was performed at 95 °C for 5 min, achieving a remarkable 99.99% lysis efficiency.

The pretreatment systems have advanced by integrating magnetic particles for separation and extraction, improving efficiency, reducing processing time, and prioritizing user convenience [42,43,44]. Shang et al. [45] developed a portable microfluidic biosensor specifically designed to detect *E. coli* O157. This biosensor features three functional zones dedicated to immunomagnetic separation, nucleic acid extraction and purification, and signal detection (Figure 1A). Initially, antibody-modified magnetic nanoparticles (MNPs) capture the target bacteria from the sample. Following lysis, silica-coated MNPs bind to and purify the DNA through a series of washing and elution steps. The extracted DNA then undergoes recombinase polymerase amplification and CRISPR/Cas12a-based fluorescence detection. Wang et al. [46] enhanced the DNA extraction method by developing a continuous-flow method that utilizes 3D printing technology and magnetic silica beads (Figure 1B). This innovative method allows for efficient DNA extraction (90% ≤) from large bacterial sample volumes (10 mL), seamlessly integrating with microfluidic PCR for targeted bacterial identification, thereby improving sample throughput and processing capabilities. Additionally, DNA extraction for foodborne pathogens was facilitated using a magnetic bead-based DNA extraction channel within a channel digital microfluidic platform [6]. In this setup, the lysis buffer and magnetic beads are preloaded at designated locations, and the magnetic bead carrying the DNA undergoes multiple washing cycles. The purified DNA is subsequently eluted and distributed to parallel reaction sites.

Chitosan-based nucleic acid extraction was used for sample preparation on a centrifugal microfluidic platform, as illustrated in Figure 1C [47]. The microfluidic channel was modified with chitosan to facilitate nucleic acid capture via electrostatic adsorption. Following the washing and elution steps, the nucleic acid templates were transferred to the recombinase-aided amplification (RAA)-T7-CRISPR/Cas 13a-based DNA detection system integrated on the chip. The nucleic acid extraction chip incorporated a glass microfiber filter designed to bind the released DNA from the bacterial lysate (Figure 1D) [48]. The DNA templates eluted from the extraction chip were then transferred to the downstream detection chip for recombinase polymerase amplification (RPA). Studies have indicated that the area and pore size of the glass microfiber filters significantly influence reagent retention capacity, extraction efficiency, and the concentration and quality of the DNA templates. For example, as the pore area increases (from 5 to 10 mm), more DNA can be trapped, increasing DNA concentration. However, beyond a certain size, a larger buffer volume is needed. Considering economic factors, a 9 mm pore area was selected as optimal. Smaller pores, on the other hand, enhance DNA entrapment by the glass fiber, improving retention capacity.

## 3. Amplification and Detection of Pathogens and Contaminants in Food

The extracted and purified nucleic acid samples in the LOC devices can undergo amplification and real-time detection. As the samples traverse the microfluidic channels, target DNA or RNA is amplified using methods such as PCR, LAMP, RAA, and RPA. This amplification process is followed by detection through colorimetric, fluorometric, or electrochemical sensing systems. LOC-based nucleic acid detection systems are increasingly utilized in the food industry to detect foodborne pathogens and GMOs and to differentiate between varieties and countries of origin. The recent advancements in microfluidic LOC devices for nucleic acid analysis are summarized in Table 1.

### 3.1. Detection of Foodborne Pathogens

Foodborne pathogens, including a range of bacteria and viruses, present significant health risks to humans through both consumption and contact, frequently manifesting symptoms such as high fever, abdominal pain, and diarrhea. Most foodborne illness outbreaks are caused by pathogens such as *Norovirus*, *Campylobacter*, *Salmonella*, *Listeria monocytogenes (L. monocytogenes)*, and *Shiga toxin-producing E*. *coli*. Other pathogens, including *Staphylococcus aureus (S. aureus)*, *Clostridium* species, *Bacillus cereus (B. cereus)*, *Yersinia enterocolitica*, and various parasites, can also occasionally cause illness [79]. Compared to immunology-based detection methods (detection limits: 10^2^–10^6^ CFU/mL) [80], amplification-based techniques offer significantly higher sensitivity (10^0^–10^4^ CFU/mL) for detecting foodborne pathogens [81]. However, traditional amplification methods are costly due to the need for complex thermal cycling instrumentation and skilled personnel, and some analysis takes a couple of hours. To address these critical threats to food safety and economic issues, LOC-based nucleic acid analysis technologies have emerged as promising tools for the rapid and accurate detection of these pathogens [7,52,82,83,84].

Xiang et al. [58] introduced a centrifugal microfluidic system that employs programable LAMP in conjunction with propidium monoazide for the quantitative detection of six viable foodborne pathogens, achieving detection thresholds of 10^2^–10^3^ CFU/mL without pre-enrichment and 10^1^ CFU/mL with enrichment. Another notable contribution was the establishment of a multiplex digital microfluidic-LAMP assay for the simultaneous detection of foodborne pathogens, including *S. aureus*, *Salmonella typhimurium (S. typhimurium)*, *E. coli*, and *L. monocytogenes*, which accomplished detection within 50 min and achieved a detection limit of 10^2^ CFU/mL in a 2 µL reaction volume (Figure 2C) [62]. Natsuhara et al. [66] introduced a centrifugal microfluidic device capable of simultaneously detecting four foodborne pathogens—*Salmonella* spp., *Vibrio parahaemolyticus* (*V. parahaemolyticus)*, *Campylobacter* spp., and Norovirus genogroup II—within 60 min using a colorimetric LAMP reaction while effectively preventing cross-contamination through direct liquid dispensing.

Xing et al. [85] have further contributed to this field by developing a finger-actuated microfluidic biosensor that integrates CRISPR/Cas12a with RAA for the multiplex detection of *B. cereus* and six other pathogens (Figure 2B). Under optimized conditions, this biosensor demonstrated the capability to test seven pathogenic bacteria within one hour, achieving limits of detection of less than 500 CFU/mL and recoveries ranging from 90% to 116% for spiked samples. Moreover, Xiang et al. [51] presented a high-throughput microfluidic system utilizing RAA specifically for the detection of the *Salmonella* genus and its specific serogroups, reporting assay times ranging from 15 to 40 min, with detection limits as low as 10^1^ CFU/mL, alongside noted enhancements in speed and reliability when compared to conventional methodologies.

Li et al. [68] developed a centrifugal microfluidic chip designed for the parallel detection of various pathogens in food, including those associated with acute hepatopancreatic necrosis caused by *Vibrio* spp., white spot syndrome virus, infectious hypodermal and hematopoietic necrosis virus, shrimp hemocyte iridescent virus, and *Enterocytozoon hepatopenaei* (Figure 2A). This real-time fluorogenic RPA-based chip operates with minimalsample volumes (5 µL) at 39 °C in 20 min, achieving a detection limit of 10 copies/µL. Further advancements are represented by a fully integrated micro-platform developed for the detection of multiple pathogens, including *S. typhimurium*, *S*. *aureus*, *B*. *cereus*, *Pseudomonas aeruginosa (P. aeruginosa)*, *E. coli* O157:H7, *Cronobacter sakazakii*, *L*. *monocytogenes*, and *V. parahaemolyticus*. This platform incorporates on-chip nucleic acid extraction, RPA, and signal detection [48] within a compact framework, where purified DNA templates were amplified at reduced temperatures (35–43 °C) over varying time periods (5–30 min). The resultant fluorescence images can be acquired via a smartphone, enabling complete analysis within 60 min.

### 3.2. Identification of Genetic Markers and Contaminants

LOC-based nucleic acid amplification and detection technologies are becoming increasingly prevalent for identifying species adulteration in food products, as well as for detecting specific GMO genes and allergens.

Zhang et al. [71] developed a real-time fluorescent LAMP microfluidic assay capable of simultaneously detecting pork, beef, sheep, and duck in food samples. This assay utilizes species-specific primers that are precoated in the reaction chambers of the real-time LAMP chip, enabling amplification to occur within 30 min. It demonstrated high specificity and sensitivity, achieving a detection limit of 0.1% (w/w) in simulated adulterated samples, in accordance with industry authentication standards. Similarly, Xiao et al. [86] introduced a hand-held microfluidic chip for on-site species authentication in meat samples (Figure 3A). This innovative system combines microneedle DNA extraction with visual LAMP and operates simply by pricking the meat, hand-shaking the chip, and applying isothermal heating. It successfully detected six different meat species and identified adulteration levels as low as 1% within 60 min, showcasing both high specificity and user-friendliness. Furthermore, a microarray PCR-directed microfluidic lateral flow strip (LFS) device was developed for the accurate, simultaneous detection of beef adulteration with chicken, duck, and pork, particularly in processed products (Figure 3B) [72]. This microarray approach significantly enhances the precision and speed of species identification in complex meat samples. Yu et al. [69] also presented a centrifugal microfluidic chip-based real-time fluorescent multiplex LAMP assay for the simultaneous detection of milk from cow, camel, horse, goat, and yak species. With a detection limit of 2.5% for adulterated milk samples, this assay is completed in 90 min, providing a robust solution for testing dairy authenticity. Kim et al. [73] reported on-site rapid identification of six laver species with a quantitative PCR system combined with a microfluidic detection chip. With a simple DNA extraction method, species-specific primers, and the high sensitivity of the developed assay, they rapidly identified seaweed species in 79 laver samples.

In their study on GMO food analysis, Loo et al. [74] addressed the challenges of on-site GMO screening using a portable microfluidic lab-on-a-disk platform. This innovative system enables rapid sample-to-answer detection of genetically modified papaya through LAMP technology. Within just 15 min, GM papaya samples can be differentiated from non-GM samples by detecting a specific GM DNA marker alongside a non-GM DNA marker. The assay exhibits impressive detection limits of 10 pg/µL for DNA and 0.02 µL for papaya juice, thereby enhancing the efficiency of GMO testing by delivering rapid and accurate results. Moreover, a novel multiplex nucleic acid testing system was created by integrating a mini-disk capillary array with visual LAMP and rapid DNA extraction (mDC-LAMP). Its performance was evaluated with GM maize and GM rice (Figure 3C) [75]. The results indicate that mDC-LAMP provides high specificity, avoiding cross-contamination, and demonstrates excellent sensitivity, with a detection limit of 25 copies per reaction.

Natcuhara et al. reported on food allergen detection utilizing a LOC device and nucleic acid analysis [78]. They demonstrated a colorimetric LAMP assay on a compact diagnostic device for the simultaneous detection of multiple allergens, such as wheat, buckwheat, and peanuts. The assay is completed within 60 min at 60 °C, highlighting its effectiveness for monitoring allergens in food products. Bourdat et al. introduced an automated microfluidic platform for on-site allergen detection using multiplex qPCR [87]. The system was integrated into a cartridge for DNA extraction, purification, and detection with an instrument featuring pneumatic, thermal, and optical systems. It detected four allergens—gluten, sesame, soy, and hazelnut—in complex food samples within two hours. Validation controls ensured accuracy, and the platform met regulatory thresholds, including 20 ppm for gluten, validated against ELISA. Ma et al. combined PCR-based genetic detection with microfluidics for peanut DNA analysis [88]. The device generated emulsion droplets, reducing reagent evaporation by 7.24%, stabilizing fluid flow, and improving PCR efficiency compared to single-phase microfluidic chips. PCR performance was validated by comparing peanut DNA detection with a commercial PCR thermal cycler, showing increased fluorescence intensity using SYBR Green. Additionally, the chip successfully amplified DNA from sesame, *Salmonella* spp., and *S. aureus*, highlighting its versatility and reliability.

## 4. Integrated Platform for Food Safety and Quality Control Applications

The miniaturization, automation, and portability of LOC devices enable real-time monitoring and on-site detection of pathogens and contaminants in food supply chains. This capability has proven invaluable for food safety applications throughout distribution channels, import inspection points, and production sites, where timely, on-site analysis is crucial. Notably, recent advancements in integrated LOC devices have consolidated DNA extraction, amplification, and detection into a single chip, progressing toward meeting industry demands for efficient and rapid food safety testing (Table 2).

Yin et al. [42] described a “sample-in-multiplex-digital-answer-out” chip that integrates DNA extraction, multiplex digital RPA, and fluorescence detection (Figure 4A). This system facilitates instrument-independent, magnetic bead-based nucleic acid extraction in just 15 min and can detect three bacterial pathogens with a limit of 10 cells per pathogen in contaminated milk within 45 min. Wu et al. [89] developed a microfluidic chip combined with RAA for rapid *S. typhimurium* (Figure 4B). This chip employs a noncontact eddy heater for bacterial cell lysis and a 3D-printed mixer for continuous-flow mixing, reaching a detection limit of 89 CFU/mL within 40 min. Sun et al. [91] integrated spore purification, nucleic acid release, amplification, and fluorescence detection into a single chip measuring 60 mm × 30 mm (Figure 4C). Using a micro air pump, spores are separated from impurities and collected via airflow. This chip demonstrated 100% specificity with a detection limit of 100 copies/reaction.

Xie et al. developed a comprehensive platform that integrates microbial cell lysis, DNA extraction, colorimetric LAMP, and detection (Figure 4D) [6]. This innovative system utilizes magnetic beads for DNA extraction, enabling the multiplex detection of *E. coli*, *L. monocytogenes*, *S. typhimurium*, and *S. aureus*, with sensitivities of 10^2^–10^3^ CFU/mL in spiked meat samples. The fully automated and user-friendly process can be completed within 60 min, with results visually confirmed on the chip. Additionally, Liu et al. [93] designed a portable microfluidic analyzer capable of detecting five foodborne bacteria within 70 min. This analyzer combines bacterial lysis, visible LAMP amplification, and detection on a centrifugal chip. Its compact design (151 × 134 × 110 mm) includes two motors strategically placed on either side of the centrifugal chip for optimal performance.

An innovative method for the on-chip culturing of rice false smut spores (RFSS) has been introduced to address the challenge of breaking hard walls for integrated detection (Figure 4E) [92]. The RFSS mycelium cultivated on the chip is easily lysed to release DNA, thereby eliminating the need for manual extraction and addressing the difficulties associated with wall breaking. The chip employs aerodynamic principles to capture the RFSS and facilitates gas–liquid coupling through a simple microvalve structure. Furthermore, liquid mixing is enhanced by a micromixer, enabling swift detection using RPA and lateral flow dipsticks. This chip offers a “sample-in, answer-out” approach, boasting a detection sensitivity ranging from 1 × 10^2^ to 1 × 10⁵ CFU/mL while requiring minimal instrumentation and delivering high sensitivity and specificity. Additionally, a disposable, instrument-free nucleic acid microfluidic cassette for detecting adulterated goat milk has been developed, integrating DNA extraction with LAMP and lateral flow strips (LFS) (Figure 4F) [70]. The steps for the LAMP and LFS assays are consolidated into a single microfluidic chip, facilitating easy on-site detection and reducing the risk of aerosol contamination from the LAMP products. The procedure encompasses DNA extraction, followed by the addition and incubation of short nucleic acids, yielding visual results within 50 min.

## 5. Future Trends and Emerging Technologies

Recent research on microfluidic LOC devices for the analysis of nucleic acids related to foodborne pathogens and contaminants presents efficient and rapid solutions for verifying food safety and authenticity. LOC technology significantly reduces assay times and effectively handles multiple analytes, yielding accurate and reliable results. This capability allows for timely responses to food safety concerns and helps mitigate associated risks. Additionally, when compared to traditional laboratory-based analytical techniques, compact and automated LOC devices offer cost savings and minimize environmental impact.

Despite the considerable advantages LOC devices provide for food safety applications, they also face several challenges. Detailed technical optimization is essential to ensure high accuracy when detecting and analyzing complex food samples. Given that LOC devices for nucleic acid analysis are often tailored to specific types of food, they require customized design modifications to enhance the extraction, amplification, and detection of DNA or RNA. On the other hand, developing a versatile device capable of being applied across a wide range of food types with varying physical properties and compositions could be considered [66,96]. The relatively high initial development costs of these devices also pose a barrier to their widespread adoption. Furthermore, scaling up production while maintaining performance standards is a significant challenge that limits commercialization and broader accessibility. Utilizing cost-effective materials amenable to large-scale production could contribute to significant cost reduction [5].

Future advancements are expected to prioritize the development of more accurate sensing systems and analytical methods to address current limitations. By integrating data analysis and artificial intelligence, developers can create highly effective and reliable food safety and quality control systems. Technologies like isothermal nucleic acid amplification, including LAMP and RPA, along with CRISPR-based assays, are likely to become more sophisticated, providing rapid, highly sensitive, and simple methods for LOC devices.

Digital detection technologies will facilitate precise quantification and multiplex analysis, allowing for the simultaneous detection of multiple pathogens or contaminants and generating standardized, quantitative results in minutes. As consumer demand for information regarding food safety and quality increases, these innovations are anticipated to have a direct impact on daily life. The integration of smartphones [97] and IoT (Internet of Things) connectivity is expected to significantly enhance future LOC devices by enabling continuous data collection, storage, and transmission to cloud-based or centralized databases. This connectivity will allow for more efficient tracking and analysis of food safety indicators, providing authorities and companies with real-time alerts, predictive insights, and compliance monitoring. Additionally, mobile applications capable of real-time food safety verification are expected to empower consumers to conveniently assess food quality and safety.

LOC devices designed for nucleic acid testing will prioritize portability, user-friendliness, and field deployability, ensuring reliable results at the point of need. Fully self-powered or battery-operated LOC platforms will be essential for operation in remote areas, distribution points, and high-throughput environments. Progress in microfabrication and materials science will enhance the practicality of cost-effective LOC platforms.

## 6. Conclusions

LOC devices have become essential tools for rapid nucleic acid analysis in food safety, integrating sample preparation, nucleic acid extraction, amplification, and detection into compact, automated systems. These devices effectively address the significant limitations of traditional food safety testing by offering high accuracy, speed, and cost-efficiency; minimizing sample volumes; and enabling POCT. Utilizing microfluidic channels and structures, LOC platforms streamline the processing and testing of food samples, significantly reducing response times for detecting foodborne pathogens and contaminants. To enhance sensitivity and specificity in identifying pathogens and contaminants, DNA preparation methods utilizing electrostatic or magnetic interactions are frequently combined with molecular diagnostic techniques such as LAMP, RAA, RPA, and CRISPR-Cas12a-based detection within LOC systems. Additionally, innovations in on-chip integration have led to the creation of portable, self-powered LOC devices for conducting on-site analysis. However, challenges remain in the development of LOC devices, including the need for high technical accuracy, customization for diverse food matrices, and cost-effectiveness for commercial applications. Future trends are focused on refining LOC devices through cost-effective design, digital detection, multiplex analysis, and IoT capabilities to facilitate real-time, quantitative monitoring of food safety. With ongoing advancements, LOC devices are anticipated to significantly improve food safety management, making real-time quality assurance viable across global food supply chains.

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
