# Peer review of "Lab-on-a-Chip Devices for Nucleic Acid Analysis in Food Safety"

_micromachines, 2024, doi:10.3390/mi15121524_

Round 1
Reviewer 1 Report
Comments and Suggestions for Authors
The review summarized Lab-on-a-Chip Devices for Nucleic Acid Analysis in Food Safety. The review can be considered for publication in "Micromachines" after revising the following questions. The comments are below.
1) Although the review already includes some comparative experiments to verify the superiority of microfluidic technology, the authors are advised to further increase the diversity of comparative experiments. For instance, they could attempt to compare microfluidic technology with other emerging technologies (such as nanotechnology and biosensing technology) to more comprehensively demonstrate the advantages and limitations of microfluidic technology.
2) Although the review provides a comprehensive introduction to the application of microfluidic technology in food safety detection, it discusses relatively little about the technical bottlenecks and challenges. The author is suggested to delve deeper into the current challenges faced by microfluidic technology, such as sample processing, data analysis, and cost control, and propose possible solutions or research directions.
3) In line 27, The ways in which the LOC chip design facilitates how samples are moved, mixed, separated, and analyzed should be described.
4) In line 47, Molecular diagnostics amplify trace amounts of DNA or RNA, rapidly detect specific pathogens or harmful substances, and when completed, explain how they have greatly contributed to public health and consumer trust.
5) The structure of the paper needs revision as section 2.2 has been introduced without including Section 2.1.
6) While there is no doubt that LAMP has significantly reduced detection time compared to other conventional methods, the authors should also acknowledge its limitations and challenges such as primer design complexity, or limited quantitative capacity which would provide a balanced perspective to readers.
7) We suggest that the author elaborate and analyze more Lab-on-a-chip's work in the introduction to enhance the authoritativeness and credibility of the review. Sensors and Actuators B: Chemical, DOI: https://doi.org/10.1016/j.snb.2023.133939 Biosensors and Bioelectronics, DOI: https://doi.org/10.1016/j.bios.2024.116338
8) In the review, a picture of scheme to summarize the content of the article is recommended. It helps the reader understand the scope of the article.
Comments on the Quality of English LanguageNo
Author Response
Responses to Reviewer 1 comments:
Reviewer #1
The review summarized Lab-on-a-Chip Devices for Nucleic Acid Analysis in Food Safety. The review can be considered for publication in "Micromachines" after revising the following questions. The comments are below.
Comment 1) Although the review already includes some comparative experiments to verify the superiority of microfluidic technology, the authors are advised to further increase the diversity of comparative experiments. For instance, they could attempt to compare microfluidic technology with other emerging technologies (such as nanotechnology and biosensing technology) to more comprehensively demonstrate the advantages and limitations of microfluidic technology.
Response to Comment 1) We appreciate the reviewer’s invaluable suggestions. In response, we mentioned integrating nanomaterials and biosensing technology into nucleic acid analysis in the revised manuscript.
In line 78-85: Integrating nanomaterials and biosensing technologies into each step of nucleic acid analysis has been widely reported (Wang, Li et al. 2020, Arshad, Deliorman et al. 2023, Xue, Jiang et al. 2023). These innovations have significantly enhanced the sensitivity and accuracy of nucleic acid analysis while improving portability and user convenience, representing substantial progress in the field. Challenges in high-performance nano-biosensor technology include simultaneously quantifying multiple biomarkers and advancing novel materials, amplification strategies, signal probes, and surface engineering, particularly optimizing surface blocking and washing steps (Bonanni and del Valle 2010, Sharma, Ragavan et al. 2015, Vikrant, Bhardwaj et al. 2019).
References:
- Wang, Z.-y., P. Li, L. Cui, J.-G. Qiu, B. Jiang and C.-y. Zhang (2020). "Integration of nanomaterials with nucleic acid amplification approaches for biosensing." TrAC Trends in Analytical Chemistry 129: 115959.
- Arshad, F., M. Deliorman, P. Sukumar, M. A. Qasaimeh, J. S. Olarve, G. N. Santos, V. Bansal and M. U. Ahmed (2023). "Recent developments and applications of nanomaterial-based lab-on-a-chip devices for sustainable agri-food industries." Trends in Food Science & Technology 136: 145-158.
- Xue, L., F. Jiang, X. Xi, Y. Li and J. Lin (2023). "Lab-on-chip separation and biosensing of pathogens in agri-food." Trends in Food Science & Technology 137: 92-103.
- Bonanni, A. and M. del Valle (2010). "Use of nanomaterials for impedimetric DNA sensors: A review." Analytica Chimica Acta 678(1): 7-17.
- Sharma, R., K. V. Ragavan, M. S. Thakur and K. S. M. S. Raghavarao (2015). "Recent advances in nanoparticle based aptasensors for food contaminants." Biosensors and Bioelectronics 74: 612-627.
- Vikrant, K., N. Bhardwaj, S. K. Bhardwaj, K.-H. Kim and A. Deep (2019). "Nanomaterials as efficient platforms for sensing DNA." Biomaterials 214: 119215.
Comment 2) Although the review provides a comprehensive introduction to the application of microfluidic technology in food safety detection, it discusses relatively little about the technical bottlenecks and challenges. The author is suggested to delve deeper into the current challenges faced by microfluidic technology, such as sample processing, data analysis, and cost control, and propose possible solutions or research directions.
Response to Comment 2) Thank you for reviewer’s comment. The technical issues of the currently developed LOC devices and future research directions for addressing these challenges, as mentioned by the reviewer, have been discussed in sections 5 and 6 in the revised manuscript as follows.
In section 5: Despite the considerable advantages LOC devices provide for food safety applications, they also face several challenges. Detailed technical optimization is essential to ensure high accuracy when detecting and analyzing complex food samples. Given that LOC devices for nucleic acid analysis are often tailored to specific types of food, they require customized design modifications to enhance the extraction, amplification, and detection of DNA or RNA. On the other hand, developing a versatile device capable of being applied across a wide range of food types with varying physical properties and compositions could be considered (Jin, Ding et al. 2020, Natsuhara, Kiba et al. 2024). The relatively high initial development costs of these devices also pose a barrier to their widespread adoption. Furthermore, scaling up production while maintaining performance standards is a significant challenge that limits commercialization and broader accessibility. Utilizing cost-effective materials amenable to large-scale production could contribute to significant cost reduction(Zhou, Dou et al. 2021).
In section 6: However, challenges remain in the development of LOC devices, including the need for high technical accuracy, customization for diverse food matrices, and cost-effectiveness for commercial applications. Future trends are focused on refining LOC devices through cost-effective design, digital detection, multiplex analysis, and IoT capabilities to facilitate real-time, quantitative monitoring of food safety.
References:
- Jin, Z., G. Ding, G. Li, G. Yang, Y. Han, N. Hao, J. Deng, Y. Zhang, W. Zhang and W. Li (2020). "Rapid detection of foodborne bacterial pathogens using visual high‐throughput microfluidic chip." Journal of Chemical Technology & Biotechnology 95(5): 1460-1466.
- Natsuhara, D., Y. Kiba, R. Saito, S. Okamoto, M. Nagai, Y. Yamauchi, M. Kitamura and T. Shibata (2024). "A sequential liquid dispensing method in a centrifugal microfluidic device operating at a constant rotational speed for the multiplexed genetic detection of foodborne pathogens." RSC Advances 14(31): 22606-22617.
- Zhou, W., M. Dou, S. S. Timilsina, F. Xu and X. Li (2021). "Recent innovations in cost-effective polymer and paper hybrid microfluidic devices." Lab on a Chip 21(14): 2658-2683.
Comment 3) In line 27, The ways in which the LOC chip design facilitates how samples are moved, mixed, separated, and analyzed should be described.
Response to Comment 3) Thank you for reviewer’s valuable comment. The following content has been added to complement and enhance the relevant section:
In line 28-34: Specifically, microchambers or microchannels within a single chip are designated to perform individual functions, with fluid control relying on either passive method (e.g., capillary action, gravity, osmosis) or active systems (e.g., pressure-driven, centrifugal flows) (Iakovlev, Erofeev et al. 2022). Passive systems are characterized by their cost-effectiveness, autonomy, and portability, whereas active systems, which incorporate miniaturized mechanical components such as electromagnetics, pumps and valves, provide enhanced control and reliability throughout multiple operational steps (Arshad, Deliorman et al. 2023).
References:
- Iakovlev, A. P., A. S. Erofeev and P. V. Gorelkin (2022). "Novel Pumping Methods for Microfluidic Devices: A Comprehensive Review." Biosensors 12(11): 956.
- Arshad, F., M. Deliorman, P. Sukumar, M. A. Qasaimeh, J. S. Olarve, G. N. Santos, V. Bansal and M. U. Ahmed (2023). "Recent developments and applications of nanomaterial-based lab-on-a-chip devices for sustainable agri-food industries." Trends in Food Science & Technology 136: 145-158.
Comment 4) In line 47, Molecular diagnostics amplify trace amounts of DNA or RNA, rapidly detect specific pathogens or harmful substances, and when completed, explain how they have greatly contributed to public health and consumer trust.
Response to Comment 4) Thank you for the reviewer’s comment. The content has been revised as follows:
In line 56-58: It facilitates prompt responses to potential risks and play a critical role in preventing their spread, thereby making substantial contributions to public health and consumer trust (Sritong, Sala de Medeiros et al. 2023).
References:
- Sritong, N., M. Sala de Medeiros, L. A. Basing and J. C. Linnes (2023). "Promise and perils of paper-based point-of-care nucleic acid detection for endemic and pandemic pathogens." Lab on a Chip 23(5): 888-912.
Comment 5) The structure of the paper needs revision as section 2.2 has been introduced without including Section 2.1.
Response to Comment 5) We appreciate the reviewer’s comment. Section 2.2 has been revised to Section ‘2’.
Comment 6) While there is no doubt that LAMP has significantly reduced detection time compared to other conventional methods, the authors should also acknowledge its limitations and challenges such as primer design complexity, or limited quantitative capacity which would provide a balanced perspective to readers.
Response to Comment 6) Thank you for your valuable comment. We agree that while LAMP offers significant advantages in terms of rapid detection, it is important to recognize its limitations. Primer design complexity is indeed a challenge, as the specificity required to avoid non-specific binding can complicate the development of effective assays. We have added these limitations to the revised manuscript.
In line 94-96: However, certain limitations in LAMP technology remain, such as primer design complexity and non-specific amplification (Moehling, Choi et al. 2021).
References:
- Moehling, T. J., G. Choi, L. C. Dugan, M. Salit and R. J. Meagher (2021). "LAMP Diagnostics at the Point-of-Care: Emerging Trends and Perspectives for the Developer Community." Expert Review of Molecular Diagnostics 21(1): 43-61.
Comment 7) We suggest that the author elaborate and analyze more Lab-on-a-chip's work in the introduction to enhance the authoritativeness and credibility of the review. Sensors and Actuators B: Chemical, DOI: https://doi.org/10.1016/j.snb.2023.133939 Biosensors and Bioelectronics, DOI: https://doi.org/10.1016/j.bios.2024.116338
Response to Comment 7) We appreciate your recommendation to expand on the work related to LOC technologies in the introduction. We agree that providing a more detailed analysis of LOC systems will enhance the review's depth and credibility. We have incorporated a discussion of suggested study (Biosensors and Bioelectronics, 2024, DOI: 10.1016/j.bios.2024.116338), as one of the examples to provide readers with a more comprehensive understanding of the field insightful feedback Another study (Sensors and Actuators B: Chemical, 2023, DOI: https://doi.org/10.1016/j.snb.2023.133939) was added the one of the references for application in clinical diagnostics (Line 42)
In line 42-46: For example, a micro-automated microfluidic device was developed for on-site and rapid POCT of Escherichia coli (E. coli) O157:H7(Yin, Zhu et al. 2024). The device was connected to a mobile phone to capture RGB data via colorimetry and fluorescence channels, allowing for precise measurement of E. coli concentration.
References:
- Yin, B., H. Zhu, S. Zeng, A. S. M. M. F. Sohan, X. Wan, J. Liu, P. Zhang and X. Lin (2024). "Chip-based automated equipment for dual-mode point-of-care testing foodborne pathogens." Biosensors and Bioelectronics 257.
Comment 8) In the review, a picture of scheme to summarize the content of the article is recommended. It helps the reader understand the scope of the article.
Response to Comment 8) Thank you for the suggestion. We created and incorporated a concise and informative figure to address this recommendation. This will provide readers with a clear visual representation of the techniques and their relevance to food safety, enhancing the manuscript's clarity and impact.
Scheme 1. Schematic overview for microfluidic lab-on-a chip technology for nucleic acid analysis in food safety control.
Reviewer 2 Report
Comments and Suggestions for Authors
Title: Lab-on-a-Chip Devices for Nucleic Acid Analysis in Food Safety
This manuscript reviews Lab-on-a-Chip (LOC) devices for nucleic acid analysis, including preparation, detection, and integration. The topic is both impactful and significant, particularly in integrating LOC techniques with nucleic acid analysis for food safety. However, the manuscript lacks clarity in explaining how LOC techniques advance nucleic acid analysis in food safety and does not comprehensively summarize current LOC methods for this application. To enhance the manuscript, the authors should provide more quantitative details and conduct a more thorough review. Below are specific comments and suggestions for improvement:
1. It is recommended to include an overview figure to visually summarize the LOC techniques for nucleic acid analysis and their applications in food safety.
2. In the introduction, briefly describe the current nucleic acid extraction methods, explaining their working principles, benefits, and limitations.
3. The introduction states that LAMP technology in LOC systems delivers results significantly faster than traditional methods. To make this statement more convincing, provide a quantitative comparison, such as specific time savings or efficiency improvements.
4. Section 2.1 is missing.
5. In Section 2.2, the authors mentioned that pretreatment process for food testing are typically time consuming and labor-intensive. Please include what the current method for the pretreatment process is.
6. In section 2.2, the authors mentioned that “Wang et al. [30] enhanced the DNA extraction method by developing a continuous-flow method that utilizes 3D printing technology and magnetic silica bead. This innovative method allows for efficient DNA extraction from large bacterial sample volumes, seamlessly integrating with microfluidic PCR for targeted bacterial identification, thereby improving sample throughput and processing capabilities.” Please explain clearer on how this LOC device enhance the DNA extraction and provide a quantitative data for throughput and processing capabilities.
7. In section 2.2, chitosan based nucleic acid extraction “Studies have indicated that the area and pore size of the glass microfiber filters significantly influence reagent retention capacity, extraction efficiency, and the concentration and quality of the DNA templates.” Instead of general discussion, if the author can provide examples to show how the pore size influences these important factors, it will make the discussion more informative.
8. In section 3.1, the logic of categorization in this section is unclear. Reorganize it to present a coherent structure. If this topic focus on the detection of food borne pathogens, please explain more about detection method and how the LOC device can enhance the accurate and rapid detection. For instance, it can be colorimetric analysis, multiplex digital microfluidic analysis. electrochemical analysis etc.
9. In section 3.1, provide detailed information about the six foodborne pathogens mentioned and discuss the detection limits of conventional methodologies.
10. In section 3.2, last paragraph, the authors mentioned the food allergen detection. However, there only one work is introduced here. Expand this section by including additional studies on LOC devices combined with nucleic acid analysis for allergen detection. This will make the review more comprehensive.
Author Response
Responses to Reviewer 2 comments:
Reviewer #2
This manuscript reviews Lab-on-a-Chip (LOC) devices for nucleic acid analysis, including preparation, detection, and integration. The topic is both impactful and significant, particularly in integrating LOC techniques with nucleic acid analysis for food safety. However, the manuscript lacks clarity in explaining how LOC techniques advance nucleic acid analysis in food safety and does not comprehensively summarize current LOC methods for this application. To enhance the manuscript, the authors should provide more quantitative details and conduct a more thorough review. Below are specific comments and suggestions for improvement:
Comment 1) It is recommended to include an overview figure to visually summarize the LOC techniques for nucleic acid analysis and their applications in food safety.
Response to Comment 1) Thank you for the suggestion. We created and incorporated a concise and informative figure to address this recommendation. This will provide readers with a clear visual representation of the techniques and their relevance to food safety, enhancing the manuscript's clarity and impact.
Scheme 1. Schematic overview for microfluidic lab-on-a chip technology for nucleic acid analysis in food safety control
Comment 2) In the introduction, briefly describe the current nucleic acid extraction methods, explaining their working principles, benefits, and limitations.
Response to Comment 2) Thank you for reviewer’s insightful comment. We agree that providing an overview of current nucleic acid extraction methods in the introduction would add valuable context to the discussion. In the revised manuscript, we included a brief description of commonly used nucleic acid extraction methods.
In line 61-73: Nucleic acid analysis typically consists of three main steps: extraction (sample preparation), amplification, and detection. The nucleic acid extraction process involves cell disruption, removal of membrane lipids and proteins, elimination of other nucleic acids, nucleic acid purification, and concentration(Goldberg 2015). This process employs both chemical methods (e.g., osmotic shock, enzymatic digestion, detergents, alkali treatment) and mechanical methods (e.g., heating, homogenization, ultrasonication, pressing, ball milling)(Ali, Rampazzo et al. 2017). Solid-phase extraction is commonly used to selectively isolate target nucleic acids from solutions via specific hydrophobic, polar, and/or ionic interactions between the solute and the sorbent(Ali, Rampazzo et al. 2017). Detergents and alkali treatments offer fast, reliable, and simple methods but have low lysis efficiency and moderate DNA purity (Birnboim and Doly 1979). Mechanical methods achieve robust lysis but require specialized equipment. Solid-phase extraction, such as silica matrices, provides high-purity DNA and is user-friendly but poses challenges in recovering small DNA fragments and is not reusable (Esser, Marx et al. 2006, Green and Sambrook 2012).
References:
- Goldberg, S. (2015). "Mechanical/physical methods of cell distribution and tissue homogenization." Methods Mol Biol 1295: 1-20.
- Ali, N., R. d. C. P. Rampazzo, A. D. T. Costa and M. A. Krieger (2017). "Current Nucleic Acid Extraction Methods and Their Implications to Point-of-Care Diagnostics." BioMed Research International 2017: 1-13.
- Birnboim, H. C. and J. Doly (1979). "A rapid alkaline extraction procedure for screening recombinant plasmid DNA." Nucleic Acids Res 7(6): 1513-1523.
- Esser, K.-H., W. H. Marx and T. Lisowsky (2006). "maxXbond: first regeneration system for DNA binding silica matrices." Nature Methods 3(1): i-ii.
- Green, M. and J. Sambrook (2012). Molecular Cloning, 4th Edn, Vol. 1, Cold Spring Harbor: Cold Spring Harbor Laboratory Press.
Comment 3) The introduction states that LAMP technology in LOC systems delivers results significantly faster than traditional methods. To make this statement more convincing, provide a quantitative comparison, such as specific time savings or efficiency improvements.
Response to Comment 3) Thank you for reviewer’s helpful comment. To strengthen the statement in the introduction, we provided a quantitative comparison highlighting the time savings and efficiency improvements offered by LAMP technology in LOC systems
In line 92-94: LAMP operating under isothermal conditions (60–65°C) to produces 106–109 copies of DNA within 30–60 minutes, outperforming thermal cycling methods, which generate 230 copies in 2–3 hours (Shang, Sun et al. 2020).
References:
- Shang, Y., J. Sun, Y. Ye, J. Zhang, Y. Zhang and X. Sun (2020). "Loop-mediated isothermal amplification-based microfluidic chip for pathogen detection." Crit Rev Food Sci Nutr 60(2): 201-224.
Comment 4) Section 2.1 is missing.
Response to Comment 4) We appreciate the reviewer’s comment. Section 2.2 has been revised to Section ‘2’.
Comment 5) In Section 2.2, the authors mentioned that pretreatment process for food testing are typically time consuming and labor-intensive. Please include what the current method for the pretreatment process is.
Response to Comment 5) Thank you for the reviewer’s comment. Following the reviewer’s suggestion, a brief introduction to the current sample preparation methods for nucleic acid analysis has been added to the revised manuscript.
In line 108-120: A sample preparation process including nucleic acid extraction and purification is typically required to analyze microorganisms or genetic markers in food. The pretreatment protocol varies based on sample type (e.g., plant, animal cells, bacteria, and virus) and the intended testing method(Wang, Jiang et al. 2023) . Nucleic acid extraction begins with cell lysis to release intracellular material (Li, Xu et al. 2023). Chemical lysis typically involves alkaline solutions or surfactants to break down the plasma membrane, with sodium dodecyl sulfate aiding in protein dissolution. Mechanical lysis applies shear stress to physically disrupt the membrane, while thermal lysis uses heat to denature cell membranes. Following extraction, purification removes inhibitors to ensure efficient amplification. Methods include centrifugation, filtration, and magnetic techniques (Wang, Jiang et al. 2023). Centrifugation separates DNA from other components by differential sedimentation, while filtration uses silica membranes to bind DNA at high salt concentrations. Magnetic beads were used to bind target DNA, which is then separated by magnetic force.
References:
- Li, Z., X. Xu, D. Wang and X. Jiang (2023). "Recent advancements in nucleic acid detection with microfluidic chip for molecular diagnostics." TrAC Trends in Analytical Chemistry 158: 116871.
- Wang, J., H. Jiang, L. Pan, X. Gu, C. Xiao, P. Liu, Y. Tang, J. Fang, X. Li and C. Lu (2023). "Rapid on-site nucleic acid testing: On-chip sample preparation, amplification, and detection, and their integration into all-in-one systems." Front Bioeng Biotechnol 11: 1020430.
Comment 6) In section 2.2, the authors mentioned that “Wang et al. [30] enhanced the DNA extraction method by developing a continuous-flow method that utilizes 3D printing technology and magnetic silica bead. This innovative method allows for efficient DNA extraction from large bacterial sample volumes, seamlessly integrating with microfluidic PCR for targeted bacterial identification, thereby improving sample throughput and processing capabilities.” Please explain clearer on how this LOC device enhance the DNA extraction and provide a quantitative data for throughput and processing capabilities.
Response to Comment 6) Thank you for reviewer’s helpful comment. To clarify the statement, we added quantitative data for the throughput and processing capabilities of the study.
In line 147-150: This innovative method allows for efficient DNA extraction (90% ≤) from large bacterial sample volume volumes (10 mL), seamlessly integrating with microfluidic PCR for targeted bacterial identification, thereby improving sample throughput and processing capabilities.
Comment 7) In section 2.2, chitosan based nucleic acid extraction “Studies have indicated that the area and pore size of the glass microfiber filters significantly influence reagent retention capacity, extraction efficiency, and the concentration and quality of the DNA templates.” Instead of general discussion, if the author can provide examples to show how the pore size influences these important factors, it will make the discussion more informative.
Response to Comment 7) We appreciate the reviewer’s invaluable suggestions. To make the discussion more informative, we provided an example of the effect of pore size and area on important factors.
In line 167-171: For example, as the pore area increases (from 5 to 10 mm), more DNA can be trapped, increasing DNA concentration. However, beyond a certain size, a larger buffer volume is needed. Considering economic factors, a 9 mm pore area was selected as optimal. Smaller pores, on the other hand, enhance DNA entrapment by the glass fiber, improving retention capacity.
Comment 8) In section 3.1, the logic of categorization in this section is unclear. Reorganize it to present a coherent structure. If this topic focus on the detection of food borne pathogens, please explain more about detection method and how the LOC device can enhance the accurate and rapid detection. For instance, it can be colorimetric analysis, multiplex digital microfluidic analysis. electrochemical analysis etc.
Response to Comment 8) Thank you for reviewer’s insightful comment. We reorganized Section 3.1 to improve the logical flow and provide a more coherent structure (Line 206-243). Most of the studies introduced in the draft manuscript were based on fluorescence analysis. Therefore, the studies in the revised manuscript are categorized according to DNA amplification methods such as LAMP, RAA, and RPA. Additionally, since the order of the referenced literature has been altered, Figure 2 has been revised accordingly, as shown below.
Figure 2. Microfluidic LOC devices for nucleic acid amplification and detection of foodborne pathogens. (A) Multiplex detection of foodborne pathogens in milk using real-time LAMP on a digital microfluidic chip. Reproduced with permission from Ref (Xie, Chen et al. 2022), (B) The multiplexed detection of foodborne pathogens based on one-pot RAA-CRISPR/Cas12a assay on finger actuated microfluidic biosensor. Reproduced with permission from Ref (Xing, Shang et al. 2023), (C) The real-time RPA microfluidic chip detection platform for several pathogenic microorganisms of the penaeid shrimp. Reproduced with permission from Ref (Li, Duan et al. 2024).
Comment 9) In section 3.1, provide detailed information about the six foodborne pathogens mentioned and discuss the detection limits of conventional methodologies.
Response to Comment 9) Thank you for reviewer’s valuable comment. We have provided detailed information about the six foodborne pathogens mentioned in Section 3.1. Additionally, we discussed the detection limits of conventional methodologies for these pathogens, highlighting their limitations in terms of sensitivity, specificity, time, and cost.
In line 184-193: Most foodborne illness outbreaks are caused by pathogens such as Norovirus, Campylobacter, Salmonella, Listeria monocytogenes, and Shiga toxin-producing Escherichia coli. Other pathogens, including Staphylococcus aureus, Clostridium species, Bacillus cereus, Yersinia enterocolitica, and various parasites, can also occasionally cause illness (Gourama 2020). Compared to immunology-based detection method (detection limits: 102-106 CFU/mL)(Kabiraz, Majumdar et al. 2023), amplification-based techniques offer significantly higher sensitivity (100-104 ≥ CFU/mL) for detecting foodborne pathogens (Oh, Wang et al. 2014). However, traditional amplification methods are costly due to the need for complex thermal cycling instrumentation and skilled personnel and some analysis takes a couple of hours.
References:
- Gourama, H. (2020). Foodborne pathogens. Food safety engineering, Springer: 25-49.
- Kabiraz, M. P., P. R. Majumdar, M. M. C. Mahmud, S. Bhowmik and A. Ali (2023). "Conventional and advanced detection techniques of foodborne pathogens: A comprehensive review." Heliyon 9(4): e15482.
- Oh, D. H., J. Wang, C.-W. Lin and X. Zhao (2014). "Advances in Rapid Detection Methods for Foodborne Pathogens." Journal of Microbiology and Biotechnology 24(3): 297-312.
Comment 10) In section 3.2, last paragraph, the authors mentioned the food allergen detection. However, there only one work is introduced here. Expand this section by including additional studies on LOC devices combined with nucleic acid analysis for allergen detection. This will make the review more comprehensive.
Response to Comment 10) We appreciate the reviewer’s invaluable suggestions. To address this, we expanded Section 3.2 by including additional studies on the application of LOC devices combined with nucleic acid analysis for food allergen detection.
In line 297-310: Bourdat et al. introduced an automated microfluidic platform for on-site allergen detection using multiplex qPCR(Bourdat, den Dulk et al. 2025). The system was integrated into a cartridge for DNA extraction, purification, and detection with an instrument featuring pneumatic, thermal, and optical systems. It detected four allergens—gluten, sesame, soy, and hazelnut—in complex food samples within two hours. Validation controls ensured accuracy, and the platform met regulatory thresholds, including 20 ppm for gluten, validated against ELISA. Ma et al. combined PCR-based genetic detection with microfluidics for peanut DNA analysis (Ma, Chiang et al. 2019). The device generated emulsion droplets, reducing reagent evaporation by 7.24%, stabilizing fluid flow, and improving PCR efficiency compared to single-phase microfluidic chips. PCR performance was validated by comparing peanut DNA detection with a commercial PCR thermal cycler, showing increased fluorescence intensity using SYBR Green. Additionally, the chip successfully amplified DNA from sesame, Salmonella spp., and Staphylococcus aureus, highlighting its versatility and reliability.
References:
- Bourdat, A.-G., R. den Dulk, B. Serrano, F. Boizot, G. Clarebout, X. Mermet, R. Charles, J. Porcherot, A. Keiser, M. Alessio, P. Laurent, N. Sarrut and M. Cubizolles (2025). "An integrated microfluidic platform for on-site qPCR analysis: food allergen detection from sample to result." Lab on a Chip.
- Ma, S.-Y., Y.-C. Chiang, C.-H. Hsu, J.-J. Chen, C.-C. Hsu, A.-C. Chao and Y.-S. Lin (2019). "Peanut Detection Using Droplet Microfluidic Polymerase Chain Reaction Device." Journal of Sensors 2019: 1-9.
Round 2
Reviewer 1 Report
Comments and Suggestions for Authors
The author has made appropriate revisions to the reviewer's comments.
Reviewer 2 Report
Comments and Suggestions for Authors
The authors address the comments by the reviewer.